# CNEURO-201, an Anti-amyloidogenic Agent and σ1-Receptor Agonist, Improves Cognition in the *3xTg* Mouse Model of Alzheimer’s Disease by Multiple Actions in the Pathology

**DOI:** 10.3390/ijms26031301

**Published:** 2025-02-03

**Authors:** Humberto Martínez-Orozco, Alberto Bencomo-Martínez, Juan Pablo Maya-Arteaga, Pedro Francisco Rubio-De Anda, Fausto Sanabria-Romero, Zyanya Gloria Mena Casas, Isaac Rodríguez-Vargas, Ana Gabriela Hernández-Puga, Marquiza Sablón-Carrazana, Roberto Menéndez-Soto del Valle, Chryslaine Rodríguez-Tanty, Sofía Díaz-Cintra

**Affiliations:** 1Departamento de Neurobiología del Desarrollo y Neurofisiología, Instituto de Neurobiología-UNAM Campus Juriquilla, Boulevard Juriquilla 3001, Juriquilla 76230, Querétaro, Mexico; h_martinez@live.com.mx (H.M.-O.); juanpablo.maya@hotmail.com (J.P.M.-A.); franciscorubiodeanda@gmail.com (P.F.R.-D.A.); fausto.sanrom@gmail.com (F.S.-R.); zyanyamc@gmail.com (Z.G.M.C.); isaacvr.1108@gmail.com (I.R.-V.); 2Departamento de Farmacología, Centro de Neurociencias de Cuba, Avenida Independencia 8126, La Habana 11600, Cuba; bencomom1985@gmail.com (A.B.-M.); marquiza@cneuro.edu.cu (M.S.-C.); roberto.menendez@cneuro.edu.cu (R.M.-S.d.V.); chris@cneuro.edu.cu (C.R.-T.); 3Centro de Investigación Biomédica Avanzada, Facultad de Medicina, Universidad Autónoma de Querétaro, Carretera a Chichimequillas S/N, Santiago de Querétaro 76140, Querétaro, Mexico; ana.gabriela.hernandez@uaq.mx

**Keywords:** multitarget drug, neurodegeneration, acetylcholine metabolism, oxidative stress, glial activation, amyloid plaque disaggregation, memory improvement, Amylovis

## Abstract

The complexity of Alzheimer’s disease (AD) pathophysiology represents a significant challenge in the development of effective therapeutic agents for its treatment. CNEURO-201 (CN, also Amylovis-201) is a novel pharmaceutical agent with dual activity as an anti-amyloid-β (Aβ) agent and σ1 receptor agonist. CN exhibits great efficacy at very low doses, delaying cognitive impairment and alleviating Aβ load in animal models of AD. However, CN functions on other remains related to this pathology remain to be investigated. The present study sought to evaluate the effects of CN treatment at a dosage of 0.1 mg kg^−1^ (p.o) over an eight-week period in the *3xTg*-AD mouse model. In silico studies, as well as biochemical and immunofluorescence assays, were conducted on brain tissue to investigate the CN effects on acetylcholine metabolism, redox system, and glial cell activation-related biomarkers in brain regions that are relevant for memory. The results demonstrated that CN effectively rescues cognitive impairment of *3xTg*-AD mice by influencing glial activity to reduce existing Aβ plaques but also modulating acetylcholine metabolism and the enzymatic response of proteins involved in the redox system. Our outcomes reinforced the potential of CN in treating AD by acting on multiple pathways altered in this disease.

## 1. Introduction

Alzheimer’s disease (AD) is a neurodegenerative disorder without a cure or effective treatment against its progression to date. Worldwide, AD is the main type of dementia (60% of cases), affecting millions of individuals of advanced age [1]. Moreover, the development of effective pharmacological therapies for AD is still an enormous challenge. One of the largest barriers to this issue lies in the complexity of AD pathophysiology, which encompasses a variety of alterations in the brain that contribute to progressive neuronal degeneration and loss of cognitive functions [2,3,4,5,6,7,8,9]. 

Research has suggested that AD etiology is related to the presence of abnormal protein deposits, which were initially described as neuritic or senile plaques [10,11] but are now recognized as amyloid plaques since they are mainly composed of amyloid-β (Aβ) peptides [11]. The central amyloid hypothesis of AD [12], which remains valid to this day [2,3,4,5,6,7,8,9,13], is based on these abnormalities. In this hypothesis, the physicochemical properties of some Aβ peptides, like Aβ_1-42_, and their aggregation dynamics allow for the formation of oligomeric species that suffer rearrangements to build fibrillar structures that become insoluble and finally are deposited as plaques in the brain tissue [14]. Therefore, AD emerges from an overproduction and accumulation of Aβ peptides that eventually propitiates the aggregation phenomena of their monomers [14], which trigger a variety of alterations, like disruption of synaptic plasticity, induction of *tau*-pathology, neuroinflammation, mitochondrial dysfunction and oxidative stress, alterations in neurotransmitter signaling, and deterioration of brain networks [15]. However, most of the clinical trials aimed at treating AD have failed [15]. Those that have received accelerated FDA approval, such as immunotherapies (Aducanumab, Lecanemab), have shown modest improvements and are associated with significant adverse events like edema and hemorrhages [16,17].

In the 1970s, histological studies in post mortem brain tissue of AD patients revealed novel insights, indicating deficiencies for enzymes like glutamate acid decarboxylase (GAD), which participates in GABA synthesis, and choline acetyltransferase (ChAT) and acetylcholinesterase (AChE), which are involved in the synthesis and degradation of acetylcholine (ACh), respectively [18,19]. Further studies in the brain of AD subjects confirmed reduced uptake and release of ACh in brain regions related to memory processing [20,21], as well as the neurodegeneration of cholinergic neurons in the basal forebrain [22]. These findings led to the hypothesis that AD was associated with dysfunction in cholinergic signaling and the subsequent proposal of AChE inhibitors as potential candidates to treat AD, under the principle that increasing the concentrations of ACh would aid the cognitive symptoms and pathological progression of AD [23]. Today, there are three AChE inhibitors available to treat AD: donepezil, rivastigmine, and galantamine. This class of drugs has shown important benefits in AD patients, alleviating cognitive performance, but adverse effects are still a challenge [24]. Nonetheless, inhibiting AChE can influence other cellular processes also involved in AD pathologies, such as immune response, glial activity, neuroprotective signaling pathways, autophagy, and Aβ deposition [25]. 

The development of drugs against AD requires a new vision that brings together the knowledge acquired so far about the disease. Therefore, the goal is to develop drugs that can act on various pathways altered by AD. In addition, it has been suggested that multitarget drugs could constitute a more effective therapeutic strategy against AD. 

The CNEURO-201 (CN, IUPAC name: methyl (2-{[4-(1-naphthylamino)-4-oxobutanoyl]amino}ethyl)dithiocarbamate) (Figure 1), also referred to as Amylovis-201, is a naphthalene derivative that belongs to novel family compounds denominated Amylovis, which were designed to target amyloid proteins in conformational diseases [26]. Initially, researchers described CN as a chaperone molecule that could inhibit the aggregation of human islet amyloid protein (hIAPP_1–37_), which is manifested in the early steps of Diabetes Mellitus [26,27]. Further in silico and in vitro experiments showed that CN possessed high affinity and stability to bind Aβ_1–42_ monomers, resulting in the inhibition of its aggregation, but also with Aβ_1–42_ fibrils, leading to the destabilization of the fibrillar architecture [28]. Moreover, Rivera-Marrero et al. [28] demonstrated that CN is a safety molecule. To prove its potential in AD, CN was orally administered to the *3xTg* mouse model of AD (*3xTg*-AD) [29] at a single dose of 1.0 mg kg^−1^ every day for 8 weeks, which resulted in a significant reduction in Aβ burden in the *subiculum* of the hippocampal formation, leading to some improvements in the long-term memory of these mice [28]. A further report revealed that treatment with 1.0 mg kg^−1^ CN significantly improved spatial memory consolidation in the streptozotocin-induced rat model of AD, which was associated with protective effects against neuronal loss in the *hippocampus* and *dentate gyrus* caused by streptozotocin [30]. Finally, a recent study confirmed that CN exhibits dual activity as a potent σ-1 receptor (σ1R) agonist and anti-amyloidogenic agent (i.e., it impedes aggregation and facilitates disaggregation of Aβ_25–35_ aggregates) with the greatest effects preventing cognitive deficits in the dizocilpine-induced mouse model of AD at very low doses (0.03–0.1 mg kg^−1^) [31].

Preclinical studies on CN indicate a strong potential as a candidate for treating AD. However, despite the fact that the action of CN as an anti-Aβ agent and σ1 receptor agonist is one of the mechanisms of its pharmacological action to improve AD-related symptoms, the impact of these effects in associated markers of the pathology remains unclear. Moreover, bearing in mind that the efficacy of CN has been observed at lower doses [31], the aim of this study was to perform an oral intervention in the *3xTg*-AD mice with 0.1 mg kg^−1^ CN to investigate its effects on learning and memory and relevant biomarkers associated with other altered pathways beyond Aβ. It is, therefore, anticipated that treatment with 0.1 mg kg^−1^ CN will exert a beneficial influence on cholinergic dysfunction, glial activation, and oxidative stress in memory substrate brain regions of *3xTg*-AD mice.

## 2. Results

### 2.1. CNEURO-201 Greatly Improves the Cognitive Functions of 3xTg-AD Mice

Cognitive performance, including learning, spatial memory, and episodic memory, was evaluated in *3xTg*-AD and wild-type (Wt) mice following the administration of 0.1 mg kg^−1^ CN or vehicle (Figure 2). We used the Morris water maze (MWM) task to evaluate learning and long-term spatial memory and the novel object recognition (NOR) test to assess episodic memory. These behavioral models were selected since spatial disorientation and impaired recognition are early symptoms of AD and, thus, can provide valuable insights into the disease [32]. 

In the acquisition phase of the MWM, the statistical analysis for latencies during training showed a significant effect for the group (F _(2, 27)_ = 5.116, *p* = 0.013) and time of training (F _(2.838, 76.63)_ = 73.44, *p* < 0.001), but the group × time of training interaction was not significant (F _(6, 81)_ = 1.747, *p* = 0.121). Nonetheless, it is noteworthy that 10-months-old *3xTg*-AD receiving the vehicle displayed higher latencies on day 4 of training (F _(2, 27)_ = 23.14, *p* < 0.001) compared with Wt mice (*p* < 0.001) and transgenic mice treated with CN (*p* < 0.001) (Figure 2A,B), thereby highlighting the beneficial effects that CN has on learning.

During the retention phase (Figure 2C,D), we detected a main effect for the group on latencies for the first entry into the target zone (location of the platform) (F _(2, 27)_ = 6.479, *p* = 0.005). The post hoc analysis revealed that the vehicle-treated *3xTg*-AD group displayed higher latencies than Wt mice (*p* = 0.057) (Figure 2C), indicating a significant disruption in long-term spatial memory in the transgenic mice. Importantly, the treatment with CN (*p* = 0.004) significantly reduced these values in the *3xTg*-AD mice (Figure 2C), demonstrating that CN greatly improved long-term spatial memory deficits. We did not find a significant effect for the treatment on platform crossings (H = 5.409, *p* = 0.067), although we observed that this parameter tended to be higher in the *3xTg*-AD group treated with CN (Figure 2D). However, when we evaluated the searching strategies to locate the target zone during retention (Figure 2E,F), we found that *3xTg*-AD mice receiving the vehicle utilized direct search-type strategies 50% less than Wt animals. Instead, this group was characterized by utilizing indirect (20% more) and non-specific (30%) search-type strategies to reach the target zone. Notably, the treatment with CN led to a significant change in the search strategy pattern of *3xTg*-AD mice (*p* < 0.05). We observed that CN intervention increased the execution of direct search-type strategies by 20% in *3xTg*-AD mice, which proportionally decreased the utilization of indirect and non-specific search strategies (Figure 2F). This suggests that CN improves long-term memory processing in this strain.

In the NOR test (Figure 2G), we found a significant main effect of the treatment on percentages of time spent in the novel object (F _(2, 32)_ = 4.358, *p* = 0.021) and discrimination indexes (F _(2, 32)_ = 4.410, *p* = 0.020). The post hoc analysis showed that vehicle-treated *3xTg*-AD mice exhibited a notable reduction in time preference percentages (*p* = 0.034) (Figure 2H) and discrimination indexes (*p* = 0.033) (Figure 2I) in comparison to Wt animals, indicating the presence of deficits in identifying novel objects. On the other hand, the percentages of time spent in the novel object (*p* = 0.050) and the discrimination indexes (*p* = 0.048) were significantly higher in the *3xTg*-AD group receiving CN than those treated with vehicle, reaching similar values to controls. Thus, our results indicate that treatment with CN substantially rescued episodic memory impairment in *3xTg*-AD mice.

### 2.2. In Silico Study: A Potential Interaction Between CNEURO-201 and Acetylcholinesterase

X-ray crystallographic analysis of AChE from *Torpedo californica* has identified the catalytic domain of the enzyme. This domain is not directly accessible from the surface of the protein; rather, it is located approximately 20 Å deep at the end of a long, narrow cavity called the catalytic throat. This structure appears to restrict the transit of substrates and products to the catalytic site. However, this is particularly noteworthy given that AChE activity is one of the fastest observed in nature [33]. The active center of this enzyme is subdivided into multiple zones that are near one another. On the one hand, a cavity is observed at the base, which is the catalytic triad (Ser 200, His 440, Glu 327). The catalytic triad forms the catalytic center, called CAS (catalytic active site), which is conserved in the structures found for this enzyme in different species. Furthermore, another unambiguous active center, designated PAS (peripheral anionic site), is situated at the aperture of the cavity and is defined by the residues Trp 279, Trp 84, and Phe 330 [33].

The AChE structure utilized in this study was identified in the Protein Data Bank (PDB) and co-crystallized with Donepezil (PDB code 1EVE). Donepezil was employed as a reference point. All simulations were conducted within a cavity that encompassed the active site of the enzyme at a depth of approximately 20 Å depth [34] (Figure 3).

Molecular docking simulations indicate that both Donepezil and CN may exhibit hydrophobic interactions with the amino acids Phe288, Phe290, Phe330, Phe331, Trp279, and Trp84, which constitute the internal catalytic pocket of AChE (Figure 2). Additionally, interactions with residues Ser200 and His440 (CAS), which influence the electrostatic attraction of the substrate to the hydrolysis site, were observed in the simulations. For CN, polar interactions, primarily hydrogen bonds (H-bonds), are predicted between the donor and acceptor atoms of the amido alkyl chain of the compound and the amino acids Asp72, Tyr121, Ser200, and His440. These potential interactions may assist in orienting the compounds in the optimal configuration within the enzyme’s catalytic region, leading to the most favorable conformation.

The score function values for each complex were calculated using the Vina program [35]. The scoring function value for the CN complex is comparable to that of the Donepezil–AChE complex, with a value of −11.7 kcal/mol and −9.1 kcal/mol, respectively. These findings lend support to the hypothesis that CN may act as an inhibitor of the AChE enzyme.

### 2.3. Molecular Dynamics Studies: Temporal Stability Evaluation of the CNEURO-201/AChE System

The rapid kinetics of AChE cannot be attributed solely to the conformation of the active site and the spatial arrangement of the amino acids involved in hydrolysis; rather, it is also dependent on a brief opening of a short channel at the bottom of the pocket, which has been termed the “back door”. The opening of the channel has been demonstrated through molecular dynamics studies of AChE in the presence of water [36,37,38]. These studies indicate that the channel formed allows the by-products of ACh degradation to pass through a thin active site wall in the vicinity of the amino acids Tyr442 and Trp84 [36,37,38].

We conducted molecular dynamics (MD) simulations to predict the stability of the formed complexes under neutral and aqueous loading conditions over a 200 ns timescale (Appendix A) and to assess the potential for the “back door” to open (Figure 4). The initial structures for the simulations were obtained from molecular docking simulations of AChE-CN and AChE–Donepezil complexes.

According to MD, the amino acid (a.a.) of the enzyme interacts with the ligands, leading to a structural change due to an opening in the molecular structure of AChE near Tyr 442 and Trp 84 (a.a. marked in red), which aligns with studies by Xu et al. [39] and Cheng et al. [40]. Thus, at the onset of the dynamics, the position of these a.a. allows an H-bond (3.4 Å) between them, and at the end of the simulation, this distance (between Tyr 442 Oζ and Trp 84 Nε1) varies considerably for the two complexes (Figure 4, cyan right). This variation results from the rotation of the Tyr 442 residue and the separation of the molecular loop containing Trp 84, which in turn leads to the breaking of the H-bond and, thus, the opening of the “back door.”

During the process of evolution, we estimated the temporal stability by evaluating the Root-Mean-Square Deviation (RMSD) of spatial positions and the Coulomb and Lennard-Jones energies of the attractive and repulsive forces involved in forming the complexes within the region (Appendix A). According to calculations, CN reaches stabilization at the catalytic site at 10 ns, even when the enzyme starts to vary its structure for the formation of the “back door”. The main interactions that stabilize the formation of CN-AChE complexes are of the Coulombic and van der Waals types. Throughout the MD, strongly hydrophobic interactions prevail with the aromatic pocket of the cavity (Phe 330 and Phe 331), which is reflected in the high value of the contact percentages (all above 70%), as well as with Trp 84, which is associated with the opening of the “back door” (Appendix A).

CN interacts with the a.a. Ser 200 and His 440, which are part of the CAS catalytic triad, potentially enhancing the stability of the complex formed. The obtained results suggest that CN is thought to bind to the enzyme, which could inhibit acetylcholine catalysis.

### 2.4. CNEURO-201 Reduces Acetylcholinesterase Activity and Acetylcholine Decay in the Hippocampus and Cortex of 3xTg-AD Mice

The levels of ACh are diminished in the brains of patients diagnosed with AD [7,41,42]. This reduction in ACh levels at least partially explains the dysregulation of AChE [43,44] and ChAT [45,46]. These enzymes are responsible for the hydrolysis and synthesis of ACh, respectively. 

In light of the aforementioned evidence, we conducted determinations to evaluate ACh metabolism in the mouse brain. First, we evaluated AChE activity in the *hippocampus* and cortex of *3xTg*-AD and Wt mice (Figure 5A,B). Our results demonstrated that the treatment decreased AChE activity in the *hippocampus* (H = 7.62, *p* = 0.013), which is not the case for the cortex (H = 5.571, *p* = 0.055). These effects were primarily attributed to CN, which reduced by ~42% the AChE activity in the *hippocampus* and cortex. However, only AChE activity in the *hippocampus* of CN-treated animals was significantly lower compared with Wt mice (*p* = 0.017) (Figure 5A).

Subsequently, we evaluated whether these alterations in AChE activity might have induced changes in ACh concentrations. To address this question, we determined the levels of ACh in pooled samples of mouse *hippocampus* and cortex since both structures exhibited a similar pattern of change in AChE activity depending on the group. The statistical analysis revealed a low main effect of treatments but a significant one on ACh concentrations in the brain tissue (H = 5.84, *p* = 0.046). In comparison to the Wt group, the *3xTg*-AD mice also treated with vehicle exhibited a reduction in the concentrations of ACh within the hippocampus and cortex (*p* = 0.049) (Figure 5C). However, such differences were not significant in the *3xTg*-AD after the treatment with CN (Figure 5C).

Finally, we investigated if prevention in the decay of ACh concentrations in the *hippocampus* and cortex of *3xTg*-AD mice could be associated with an increased synthesis of this neurotransmitter by ChAT. However, we did not find a significant main effect of the group on ChAT activities of both regions (*hippocampus*, H = 0.18, *p* = 0.925; cortex, H = 1.63, *p* = 0.469), suggesting no changes in the activity of this enzyme between groups (Figure 5D,E).

### 2.5. CNEURO-201 Influences Antioxidant Enzymatic Activity in the Hippocampus of 3xTg-AD Mice

Oxidative stress in the brain is a common consequence of the aging process, and it is frequently observed to be intensified in individuals with AD [47,48]. Consequently, we evaluated the levels of reactive oxygen/nitrogen species (RONS) in the *hippocampus* of Wt and *3xTg*-AD mice by measuring the fluorescence generated by the reaction of RONS with 2,7-dichlorofluorescein (DCF). The *hippocampus* was chosen because this brain structure in the *3xTg*-AD mice is highly affected by amyloid pathology. However, the results of this assay demonstrated no statistically significant change in RONS levels between the groups (H = 2.780, *p* = 0.265) (Figure 6A).

Subsequently, we investigated whether CN exerted an effect on the enzymatic activity of glutathione peroxidase (GPx) and glutathione reductase (GR), two important antioxidant enzymes [49]. A significant main effect of the group was revealed for GPx activity (H = 6.773, *p* = 0.024) in the *hippocampus*. We found that GPx activity was significantly increased (*p* = 0.045) in transgenic mice receiving CN compared with the Wt group (Figure 6B). For GR activity, the statistical analysis revealed a significant main effect for the group, which was almost significant (H = 5.655, *p* = 0.052). GR activity in the hippocampus of *3xTg*-AD mice treated with CN tended to be lower compared with Wt animals (Figure 6C).

Finally, we determined the total superoxide dismutase (SOD) enzymatic activity involved in the neutralization of O_2_^•^ radicals and the formation of H_2_O_2_ (Figure 6D). We observed a significant main effect for the group on total SOD activity (H = 7.491, *p* = 0.014). The total SOD inhibitory activity against O_2_^•^ radicals was significantly reduced in the *hippocampus* of *3xTg*-AD mice treated with vehicle compared with the Wt group (*p* = 0.025), but this difference was no longer significant for the transgenic group receiving CN (Figure 6D). 

### 2.6. CNEURO-201 Decreases Expansion and Promotes Dissagregation of Amyloid Plaques Together with Reductions in Glial Activation Markers in the Subiculum of Aged 3xTg-AD Mice

The antiaggregatory properties of Aβ by CN have been demonstrated previously in *3xTg*-AD mice [28]. Moreover, CN also promotes the disaggregation of Aβ aggregates in vitro [31]. Nevertheless, it remains unclear whether CN intervention could facilitate the disaggregation of preexisting Aβ plaques in vivo.

To this end, we designed an experiment in which we administered 0.1 mg kg^−1^ CN for eight weeks to aged *3xTg*-AD mice (12-14 months of age; n = 3-4), which are known to exhibit the presence of plaques [29,49]. We conducted the study using histological staining of brain sections with the fluorescent dye thioflavin-S, which stains dense plaques and some *tau* aggregates in the CA1 subregion [50]. To avoid bias by the possible presence of thioflavine-positive tau aggregates, only the *subiculum* of the hippocampal formation was selected as the structure of interest due to its higher Aβ plaque abundance [29,49]. The results of the experiment (Figure 7A–E) demonstrated that 0.1 mg kg^−1^ CN treatment led to a notable reduction in thioflavin+ plaques in comparison to the 3xTg-V group (*p* = 0.044) (Figure 7C). Moreover, we also detected a significant reduction in the thioflavin+ plaques area present in the *subiculum* of *3xTg*-AD mice treated with our drug (*p* < 0.001) (Figure 7D). These results corroborated not only the anti-aggregating capacity of CN but also, importantly, demonstrated the disaggregating effect of CN on existing Aβ plaques (Figure 7E).

Dense plaques mainly contain Aβ peptides but also other elements such as glial cells [11]. Both microglia and astrocytes migrate to plaque sites where they accumulate [51,52], and their response represents one of the immediate mechanisms to limit amyloid plaque expansion [53]. Moreover, these cells are highly involved in the activation of inflammatory processes because of the deposition of Aβ plaques in AD [54]. Therefore, we verified whether CN influences microglial and astrocytic activity in *3xTg*-AD mice. Thus, we performed immunofluorescence experiments to label microglia (anti-Iba-1), astrocytes (anti-GFAP), and Aβ species (anti-BAM10, which has an affinity for peptides Aβ_1-40_ and Aβ_1-42_, oligomers, fibrillar species and plaques) in the brain of *3xTg*-AD mice treated with vehicle and CN (Figure 7F,G,J,K,O,P). Our results showed that 0.1 mg/kg CN significantly reduced the Aβ-labeled area in the *subiculum* compared with the group receiving vehicle (*p* < 0.05) (Figure 7H,L). Similarly, *3xTg*-AD mice treated with CN exhibited lower labeled areas for glial fibrillary acidic protein (GFAP) (*p* = 0.017) and ionized calcium-binding adapter molecule 1 (Iba-1) (*p* = 0.019) in the same region compared with the vehicle group (Figure 7I,M). 

The present findings were consistent, which demonstrated that CN behaves as an Aβ anti-aggregating and disaggregating molecule. Importantly, our outcomes also indicate that 0.1 mg kg^−1^ CN may alleviate neuroinflammation by glial activation and plausibly suggest a role for the drug in impeding Aβ plaque growth, expansion, and elimination through direct effects on glial cells (Figure 7N). 

## 3. Discussion

The development of effective pharmacological therapies against AD is still a huge challenge. Today, it is well recognized that AD is a multifactorial disease with a highly complex physiopathology. With all this knowledge, enumerable therapeutical targets have been proposed to delay AD progression. These targets include Aβ, *tau*, growth factors and hormones, neurotransmitter receptors, and several markers for metabolism/bioenergetics, inflammation/immunity, neurogenesis, synaptic plasticity/neuroprotection, proteostasis/proteinopathies, and circadian rhythm [55]. Over the past several decades, AD drug development has been focused on molecules with a single target, such as Aβ, AChE, or N-methyl-D-aspartate receptors. These are the most recent therapies available [56]. In part, this strategy is still preferred based on financial reports showing that AD drugs targeting Aβ and circuits and synapses have more probability of completing all clinical phases and reaching commercialization [55]. Notwithstanding these significant advancements, AD persists as a global health concern, impacting millions of individuals and giving rise to considerable socio-economic consequences in most countries.

The complexity of AD pathology itself represents the most important barrier to the development of pharmacological therapies. This issue underscores the imperative for a shift in recent strategies for drug development, moving from molecules that target a single disease hallmark to multifunctional molecules that act on distinct altered pathways in AD. Particularly, σ1R has emerged as an interesting therapeutical target for these purposes since its stimulation is crucial for multiple neuroprotective pathways [57,58,59]. The σ1R was proposed by Martin et al. [60] and years later defined by Su et al. [61]. The σ1R is a molecular chaperone bound and regulated in the endoplasmic reticulum (ER) of cells [57] that is highly expressed in the brain in all neuronal cell types, including excitatory and inhibitory neurons, oligodendrocytes, astrocytes, and microglia [62,63]. Concretely, in neurodegenerative diseases like AD, the σ1R may play a biological role in the elimination of misfolded or unfolded proteins in the ER, calcium homeostasis, oxidative stress, autophagy, anti-inflammatory response, and neuroprotective mechanisms [59]. Molecules with σ1R agonist activity can contribute to NMDA receptor signaling, synaptic plasticity, and axonal growth [64], giving them the ability to alleviate learning and memory deficits [58]. 

Importantly, σ1R agonists have been shown to improve cognitive function at doses that are similar to or even lower than those used to treat AD with commercial AChE inhibitors [65]. As an example, the novel drugs ANAVEX® 3–71 and ANAVEX® 2–73, which possess dual agonist properties on both M1 and σ1R muscarinic receptors, have demonstrated efficacy in alleviating cognitive deficits, synaptic impairment, amyloid and *tau* pathologies, and neuroinflammation in murine models of AD [66,67,68,69]. Moreover, ANAVEX® 3–71 has successfully completed Phase 1 clinical trials, while ANAVEX® 2–73 is still in Phase 2b/3. As previously stated, this is merely an illustration of the potential efficacy of multifunctional drugs in the treatment of AD, specifically those with agonistic activity on σ1R.

In light of these considerations, the dual action of CN as an Aβ anti-aggregant and σ1R agonist [28,31] offers compelling evidence that this new drug could be used as a treatment for AD. We previously showed that CN oral intervention in *3xTg*-AD at a dose of 1.0 mg kg^−1^ greatly reduced the Aβ load in the hippocampal formation, which contributed to improvements in memory of this strain [28]. Consistently, here we demonstrated that CN acts as an effective anti-aggregating agent of Aβ at very low doses, resulting in an important recovery of the spatial and episodic memory of *3xTg*-AD mice. This is also consistent with recent evidence demonstrating that 0.1 and even 0.03 mg kg^−1^ CN improves memory impairment induced by dizocilpine [31]. This was interesting since drugs with σ1R agonistic activity, like ANAVEX 2-73, showed greater improvements in memory with higher doses (0.3 and 1.0 mg/kg) in the same animal model [66]. Moreover, our data agree with previous evidence supporting the potential role of σ1R agonists in preventing memory deficits associated with Aβ [66,67]. However, CN stands out from other compounds with σ1R agonistic activity for its direct interaction of CN with Aβ aggregates and further disaggregation of this species. Therefore, we determined that the therapeutic effects of low doses of CN in AD stem from an orchestrated activity between the inhibition of Aβ aggregation (anti-amyloidogenic property) and σ1R agonist feature that allow the activation of natural homeostatic mechanisms in the brain tissue. Thus, CN can display a notable capability to improve other hallmarks of AD pathology, such as increased oxidative and cholinergic dysfunction and disruption of glial processes involved in repair and anti-inflammatory responses.

An important finding of this study is that CN can eliminate existing Aβ plaques in vivo, even when it is administered at very low doses, which is consistent with recent in vitro findings [31]. Nonetheless, based on our histological analysis, which showed that Aβ burden and plaque size decreased together with glial markers, it seems that the activity of CN in vivo may be closely related to a direct action on glial cells. The precise nature of this event remains unknown; however, such behavior may be explained by the agonist effects of the drug on σ1R. This receptor is highly expressed in both astrocytes and microglia [62,63]. One of the main functions associated with σ1R stimulation is the regulation of intracellular calcium flux [70,71]. In glial cells, intracellular calcium influx modulates glial cell activities such as migration and those related to Aβ clearance [72]. Therefore, we strongly believe that CN may induce plaque disaggregation not only by its disaggregating action but also by improving σ1R-mediated glial migration and phagocytosis. Both effects acting together might improve the global beneficial action of CN at such a low dose.

Previous evidence suggests that hippocampal neurons display deficiencies in the antioxidant system, thereby reducing its efficacy in attenuating ROS formation and oxidative stress. For instance, a previous in vitro study demonstrated that hippocampal neurons isolated from 11-month-old *3xTg*-AD mice exhibited elevated levels of ROS in the media compared with neurons isolated from Wt mice of the same age [73]. In line with this, a recent in vivo study has also reported increased O_2_^•^ radicals by using the dihydroethidium method in the *hippocampus* of middle-aged *3xTg*-AD mice (7–9 months) [74]. The findings of this study contradict the evidence presented previously, as our *3xTg*-AD mice did not exhibit a concrete increase in RONS in their *hippocampus*. This suggested that the antioxidant defense system in our mice could be counteracting oxidative stress in this brain region. It is important to note that CN increases SOD activity that is reduced in the *hippocampus* of *3xTg*-AD mice, thereby facilitating the neutralization of O_2_^•^ radicals. Sequentially, GPx activity is increased in the same region, possibly to improve the H_2_O_2_ resultant by higher SOD activity. Therefore, following CN intervention, brain oxidative stress could be better handled. In addition, it is believed that stimulation of σ1R signaling promotes the expression of nuclear factor erythroid 2-related factor 2 (Nfr2) and antioxidant enzymes such as GPx, as well as reduced glutathione synthesis [59]. In this context, antioxidant response in AD pathology could be mediated by CN-induced σ1R stimulation. However, to confirm this hypothesis, further experiments will be considered to demonstrate such a hypothesis, also considering some factors like increasing the sample size or implementing other methods like aconitase or dihydroethidium method assays to detect changes of ROS in the brain tissue. 

In molecular docking and MD simulations, it has been proposed that both Donepezil and CN can interact with specific amino acids in the catalytic pocket of the AChE enzyme through hydrophobic and polar interactions facilitated by hydrogen bonds. These interactions facilitate the optimal configuration of the compounds within the catalytic region of the enzyme, thereby conferring high stability to the ligand–AChE complexes. Additionally, in both simulations, a structural alteration in the enzyme is observed because of the opening of the “back door” near Tyr 442 and Trp 84.

In accordance with this hypothesis, the administration of CN was observed to once again reduce AChE activity in the two brain regions under study, thereby preventing the decline in ACh that was evident in the vehicle-treated *3xTg*-AD mice. Although this result could be considered direct, further studies would be necessary to address the various factors influencing the cholinesterase system. In this regard, a plausible explanation may arise from the direct interaction that exists between AChE and Aβ. Previous studies have shown that AChE coexists with Aβ deposits and that the β-amyloid peptide can influence AChE levels and activity [75,76,77]. Thus, it seems logical to assume that as CN reduces Aβ burden, this may reduce AChE activity, which is of particular significance given that ChAT activity remained unaltered in both vehicle- and CN-treated *3xTg*-AD mice. This suggests that CN acts as a positive modulator of the ACh system. This result is related to a study indicating that ACh release in the brain is increased via Ca^2+^ signaling following positive σ1R stimulation [70,71]. In this context, the higher ACh concentrations found in the present study could be indicative of an increase and preservation of synaptic ACh stores rather than an increase in the synthesis of this neurotransmitter by ChAT. In light of these observations, it can be assumed that the σ1R agonist activity of CN also enhances ACh signaling.

In conclusion, the present study highlighted the potential of CN as a multifunctional drug candidate for treating AD, supporting its dual properties as an anti-Aβ agent and σ1R agonist as the main therapeutical actions. Our outcomes also extended the knowledge about the action mechanism of CN in vivo, demonstrating that CN treatment improves other hallmarks of the disease, such as oxidative stress, cholinergic dysfunction, and possibly neuroinflammation. This evidence reinforced the idea that multifunctional drugs could be more effective treatments against AD.

## 4. Materials and Methods

### 4.1. Molecular Docking

All ligand-protein molecular docking simulations were performed as described [31] using the AutoDock Vina program (Vina) [35]. The structure of the AChE enzyme was downloaded from PDB (code 1EVE).

### 4.2. Molecular Dynamics Simulations

All molecular dynamics simulations were carried out as described previously [31]. The charge of the ligand was neutral at pH 7.4, as corresponds to its structure.

The protonation states of the ionizable residues of the AChE enzyme at pH 7.4 were determined using the PDB2PQR program, which uses the PROPKA subroutine [78] to predict the functional group values of ionizable amino acids in their chemical environment. The ligand–AChE complex was minimized with 50,000 integration steps and a minimization step of 0.01 nm for 100 ps at 310 K and 1 atm.

### 4.3. Experimental Animals

This study was carried out in strict accordance with the recommendations in the Guide to the Care and Use of Laboratory Animals of the National Institutes of Health. The protocol was approved by the Ethics Committee for Animal Experimentation at the Instituto de Neurobiología (INb) of the Universidad Nacional Autónoma de México (UNAM) (Protocol Number: 117). Female Wt mice (B6129SF2/J, The Jackson Laboratory, Bar Harbor, ME, USA) and *3xTg*-AD mice (B6;129-Psen1tm1Mpm Tg(APPSwe,*tau*P301L)1Lfa/Mmjax, The Jackson laboratory) were used for this study. The ages of the mice were 8–9 or 12–14 months old. The animals were maintained in the vivarium facility of the INb in polycarbonate cages (2–4 animals per cage), provided with water and chow (Purina® 5001) ad libitum, and housed under controlled conditions (21–25 °C, 45–65% relative humidity, and a 12 h inverted dark/light cycle).

### 4.4. Pharmacological Treatments

The CN was provided by the Cuban Center of Neuroscience (98% purity) and prepared and administered to mice as previously described [28]. Briefly, the CN was finely ground in a mortar to prepare a suspension using 0.05% starch solution containing 0.05% Tween-20 as the vehicle. The suspension was kept in pharmaceutical amber glass bottles and stored at 4–8 °C. Fresh solutions were prepared every four days. Different sets of Wt and/or *3xTg*-AD mice were used in this study, and pharmacological interventions were carried out under the same conditions. Mice were divided into three groups: control group, Wt mice receiving 1.0 ml kg^−1^ vehicle; the *3xTg*-AD group receiving 1.0 ml kg^−1^ vehicle (*3xTg*-V); and the *3xTg*-AD group receiving 0.1 mg kg^−1^ CNEURO-201 (*3xTg*-CN0.1). The dose of CN was chosen because in previous studies it was shown to be one of the most effective in preventing cognitive decline in a mouse model of AD [31]. A single oral dose of the drug or vehicle was administered every day in the morning period (9:00-10:00 am) for eight weeks using a gastroesophageal curved cannula of stainless steel. After 8 weeks of treatment 10-month-old mice were subjected to behavioral tasks (one set destined for MWM and the other for NOR), whereas aged mice were euthanized, and the brain tissue was processed for the histological analyses. The treatment was maintained during the behavioral analysis period, and it was administered 40 min before starting the tasks.

### 4.5. Morris Water Maze Task

Learning and long-term spatial memory in mice were evaluated in the MWM based on a previous protocol [79]. The task was performed in a white plastic pool of 100 cm diameter and 39 cm height filled with water at 24 cm deep and a temperature between 20–23 °C. The apparatus was divided by eight cardinal points (N, NE, NW, S, SE, SW, E, y W), and an 8 × 8 cm squared platform (target zone) with a height of 23 cm was positioned 1 cm below the water level in the center of the NW quadrant (target quadrant). A light source was adjusted for appropriate illumination, and water was made opaque by adding non-toxic white paint. Visual cues were set on the walls surrounding the pool. All the sessions were recorded through a hanging camera set vertically at enough height to record the full apparatus. 

The task was performed in two phases. During the first phase (i.e., acquisition), mice were trained for four consecutive days to reach the platform and stay for at least 10 s before being rescued. On every training day, mice were subjected to four trials with a duration of a 60 s duration and a 10 min inter-trial interval between each mouse. The starting position and sequence for each trial were different every day. The time spent finding and reaching the platform (latencies) was registered for each trial to calculate average latencies per day. The animals that did not reach the platform before 60 s (latency equal to 60) were gently guided to the target, where they stayed for 10 s. The second phase (i.e., retention) was performed on the fifth day. The platform was removed, and mice were set into the pool starting at the SE position. The mice were given 60 s to find the platform. After each trial of the task (acquisition and retention), mice were gently dried with a towel and returned to their cages. The videos of the retention phase were processed using the SMART (PanLab) software version 2.5.21 to determine the time of the first entry into the target zone and the number of platform crossings. 

Additionally, tracking maps of the retention phase were processed in a non-computerized parameter-based Search Strategy Algorithm [80] to classify the type of strategies performed by mice and detect changes in spatial memory and learning. 

### 4.6. Novel Object Recognition Task 

Episodic memory was evaluated for all groups of mice through the novel object recognition (NOR) task based on a previous protocol [81]. Briefly, the task consisted of three steps with separation intervals of 24 h each: habituation (open field), familiarization (two identical objects), and test (novel object). The apparatus consisted of a cage with four compartments of 33 × 33 × 33 cm to evaluate up to four mice at the same time. Behavior was monitored and recorded through an interface to a computer. The videos were analyzed manually using a chronometer to determine the exploration times, which served to estimate the percentage of time spent on each object and discrimination index [82].

### 4.7. Tissue Isolation and Preparation

Mice were subjected to euthanasia by intraperitoneal injection of 250 mg/kg sodium pentobarbital 24 h after the Y-maze task. Deeply anesthetized animals were decapitated to obtain fresh brain samples or underwent brain fixation using intracardial perfusion with *p*-formaldehyde 4% (*m*/*v*) in 0.1 mol/L phosphate-buffered saline (PBS) pH = 7.4. 

Non-fixated brains were dissected to isolate the *hippocampus* and cortex. The samples were stored in Eppendorf® tubes, immediately frozen using solid CO_2_, and kept at −70 °C until the analysis. Brain tissue was slowly defrosted and homogenized with 600 µl (hippocampus) or 1000 µl (cortex) of lysis buffer (0.5% Triton X-100 and inhibitor proteases in PBS buffer) using a homogenizer and maintaining the sample on the ice during the procedure. Afterward, the suspensions were centrifuged for 15 min (10,000 ×g, 4 °C). Then, supernatants were collected in independent aliquots for the different biochemical analyses, immediately frozen using solid CO_2_, and kept at −70 °C until the analysis. The total protein content in the supernatants was determined by the Bradford method [83].

Fixated brains (n = 4) were used to obtain sagittal brain slices of 40 µm thickness in a cryostat for immunohistochemical analysis.

### 4.8. Acetylcholinesterase Activity 

The enzymatic activity of AChE in the hippocampus and cortex homogenates was determined using Ellman’s method [84] with some modifications for microplate assay [85]. Briefly, aliquots of frozen homogenates were slowly defrosted on ice and diluted 1:2 with 0.1 mol/L PBS (pH = 8.0). Ellman’s reagent was prepared by mixing 5,5-dithiobis(2-nitrobenzoic) acid (DTNB, 10 mmol/L; Sigma-Aldrich, St. Louis, MO, USA; #D8130) in 0.1 mol/L PBS pH = 8.0 with the same buffer in a 5:150 ratio. A volume of 138 µL of Ellman’s reagent was transferred into the wells, and 10 µL of the diluted homogenate was added for the assay. The mixture was incubated at room temperature and protected from light to allow the reaction of sulfhydryl groups in the homogenate with DTNB for 10 min. The reaction began after adding 2 µL of the substrate acetylthiocholine iodide (75 mmol/L; Sigma-Aldrich, St. Louis, MO, USA; #A5751) in 0.1 mol/L PBS pH = 8.0, and changes in absorbance at 412 nm and 25 °C were immediately measured using a microplate reader Varioskan® Lux microplate reader for 10 min at intervals of 1 min. The AChE activity was expressed as nmol of acetylthiocholine catalyzed/min and normalized per mg of protein. All the experiments were carried out in duplicate. 

### 4.9. Acetylcholine Quantification 

The quantification of ACh concentration in brain tissue was estimated indirectly using the fluorometric measurement of total and free choline with a commercial kit (Abcam, Cambridge, UK; #ab65345) following the manufacturer’s instructions. For this assay, equal volumes of *hippocampus* and cortex supernatants of each mouse were mixed individually. Then, 5 µL of the pooled sample was diluted 1:10 with the assay buffer provided in the kit and used for determination. The results were expressed in nmol and normalized per mg of protein.

### 4.10. Choline Acetyltransferase Activity 

The enzymatic activity of ChAT in the *hippocampus* and cortex homogenates was studied using the 4,4’-dithiodipyridine (4-DTP; Sigma-Aldrich, St. Louis, MO, USA; #143057) spectrophotometric method [86], with some modifications for a microassay. For the analysis, a reaction reagent was prepared by mixing 100 µL of 0.1 mol/L PBS, 10 µL of 6.2 mmol/L Acetyl coenzyme A (Sigma-Aldrich, St. Louis, MO, USA; #A2056) in 0.01 N HCl, 10 µL of 1.0 mol/L Choline chloride (Sigma-Aldrich, #C1879), 10 µL of 0.76 mmol/L Neostigmine methyl sulfate (Sigma-Aldrich, St. Louis, MO, USA; #N2126), 10 µL of 3.0 mol/L NaCl, and 10 µL of 1.1 mmol/L EDTA per each sample. Separately, 35 µL of water mixed with 15 µL of supernatant (control, denatured protein) or nothing (test) were added to 1 mL Eppendorf tubes and incubated in boiling water for 2 min. After that, a similar mixture of supernatant was added to the test tube, and 150 µL of reaction reagent preheated at 37 °C in a water bath was added to each tube. Mixtures were incubated at 37 °C in a water bath for 20 min, followed by incubation for 2 min in boiling water to stop the reaction. After cooling, 0.2 mL of water was added to each tube and mixed, and denatured protein was removed from samples by centrifugation at 12,000 rpm for 10 min. Then, 150 µL of each supernatant was collected, added to a well plate, mixed with 10 µL of 1.0 mmol/L 4-DTP, and incubated for 15 min at room temperature in the dark. The absorbances at 324 nm and 25 °C were measured using a Varioskan^®^ Lux microplate reader. The ChAT activity was expressed as µmol of acetylcholine formed/min and normalized per mg of protein considering the extinction coefficient for the 4-thiopyridine (ε_324_ = 19,800 M·L^−1^·cm^−1^). All the experiments were carried out in duplicate.

### 4.11. Determination of Reactive Oxygen/Nitrogen Species

The concentrations of RONS in the *hippocampus* and cortex homogenates were estimated through the 2,7-dichlorofluorescein diacetate (DCFH2-DA) method based on a previous protocol [87]. Before the assay, DCFH2-DA (Sigma-Aldrich, St. Louis, MO, USA; #D6883) was deacetylated with NaOH 0.1 M to obtain a 200 µM solution of DCFH2 following a previous protocol [88]. For the assay, 20 µL of supernatant from the *hippocampus* (diluted 1:1) or buffer were mixed with 140 µL of 0.1 mol/L PBS (pH = 7.4) and 40 µL of 200 µmol/L DCFH2 in the dark. Immediately, fluorescence (λ _excitation_ = 503 nm; λ _emission_ = 535 nm) for each sample and blanks were measured at 0 min and after 30 min of incubation at 37 °C using a Varioskan^®^ Lux microplate reader. The change in fluorescence was calculated in arbitrary units (RFUs) and normalized per mg of protein.

### 4.12. Glutathione Peroxydase Activity

The enzymatic activity of GPx was evaluated based on a previously described protocol [89]. Briefly, a mixture containing 5.0 mmol/L EDTA, 3.0 mM sodium azide, 1.0 mmol/L GSH (Sigma-Aldrich, St. Louis, MO, USA; #G4251), 0.14 mmol/L NADPH (Sigma-Aldrich, St. Louis, USA; #N1630), and 0.5 U/mL GR (Sigma-Aldrich, St. Louis, MO, USA; #G3664) was prepared in 0.1 mol/L PBS (pH = 7.4). A volume of 135 µL of the previous mixture, 10 µL of the homogenate samples of the *hippocampus* (diluted 1:1 in 0.1 mol/L PBS pH = 7.4) and 5 µL of 0.3 mM H_2_O_2_ were added to the wells, and decrease in the absorbance at 340 nm at 25 °C was monitored every minute for 5 min using a Varioskan^®^ Lux microplate reader. The results were expressed as nmol of NADPH oxidized/minute and normalized per mg of protein (NADPH ε_340_ = 6.200 M·L^−1^·cm^−1^).

### 4.13. Glutathione Reductase Activity

The enzymatic activity of GR was determined based on previous protocols [89,90]. Briefly, 125 µL of a mixture containing 0.5 mmol/L EDTA and 1.0 mmol/L oxidized glutathione (GSSG; Sigma-Aldrich, St. Louis, MO, USA; #G6654) in 100 mmol/L phosphate buffer pH = 7.4 was incubated with 10 µL of diluted (1:1) supernatant for 5 min at room temperature. After that, 15 µL of 1.0 mmol/L NADPH in 0.1 mol/L PBS (pH = 7.4) was added to each well to begin the reaction. The decrease in the absorbance at 340 nm at 25 °C was monitored every 30 s for 10 min using a Varioskan^®^ Lux microplate reader. The results were expressed as nmol of NADPH oxidized/minute and normalized per mg of protein (NADPH ε_340_ = 6.200 M·L^−1^·cm^−1^).

### 4.14. Total Superoxide Dismutase Activity

Total SOD activity was evaluated based on previous protocols [91,92]. Briefly, for the total SOD activity, 190 µL of a reagent containing 56 µmol/L nitroblue tetrazolium (Sigma-Aldrich, St. Louis, MO, USA; N5514), 1 mmol/L diethylenetriaminepentaacetic acid (DETAPAC; Sigma-Aldrich, St. Louis, MO, USA; #D6518), 1.0 unit Catalase (Sigma-Aldrich, #C40), 0.1 mmol/L Xanthine (Sigma-Aldrich, St. Louis, MO, USA; #X7375), 0.13 mg/mL Bovine Serum Albumin, and 0.05 mmol/L Bathocuproinedisulfonic acid (Sigma-Aldrich, St. Louis, MO, USA; #B1125) in 50 mmol/L potassium phosphate buffer pH = 7.8 was added to wells in a microplate. Aliquots of the supernatant at different protein concentrations of 0.0, 1.0, 5.0, 10.0, 25.0, 50.0, 100.0, and 150.0 µg were mixed with the reagent and incubated for 5 min at room temperature. After that, 5 µL of 0.07 U/mL Xanthine oxidase was added to begin the reaction. The change in the absorbance at 560 nm was measured every 30 s for 10 min using a Varioskan^®^ Lux microplate reader. A plot of protein concentration vs. slope was constructed, and a Michaelis–Menten fit was applied to obtain Km values (IC_50_), which served to calculate U/mg of protein (1000/Km) for each sample. The results were expressed as a percentage of control.

### 4.15. Thioflavin-S Staining

An aqueous solution containing 1% (*m*/*v*) Thioflavin S (Sigma, Cat #T-1892) was prepared and subsequently filtered before use. Sagittal brain sections previously mounted on gelatinized slides were firstly rehydrated in PBS and then immersed into 50% (*v*/*v*) ethanol and PBS once more for one minute each. After that, brain sections (n = 5–10) were incubated in 1% Thioflavin S solution for 10 min and then sequentially submerged in PBS, 50% ethanol, and PBS for 1 min each. Vectashield^®^ was used as a mounting medium. Photomicrographs in the x-y plane of the *subiculum* subregion of the hippocampal formation were obtained using an APOTOME-ZEISS fluorescence microscope with a 10X objective and a GFP filter (488 nm emission) and AxioVision software version 4.8 (Carl Zeiss, Oberkochen, Germany). The area of the fluorescence signal corresponding to Thioflavin S was determined in µm^2^ by image processing in Image J Fiji (NIH, Bethesda, MD, USA) software version 1.54 f (Java 1.8.0_322).

### 4.16. Immunofluorescence

The sagittal brain sections were processed for free-floating immunofluorescence labeling of Aβ, microglia, and astrocytes. Briefly, a total of six sagittal brain slices per mouse were subjected to an antigen retrieval procedure with citrate buffer pH = 6.0. After that, slices were incubated all night with a primary antibody solution of anti-BAM10 (1:1000, Sigma-Aldrich, A3981) and anti-GFAP (1:500; Abcam, #ab68428) or anti-Iba-1 (1:500; Abcam, #ab107159) in PBS containing 0.1% Triton-X100 and 0.5% Tween-20 at 4 °C. Then, they were incubated with their respective Alexa Fluor secondary antibodies (1:500). Tiled images in the x-y plane from the whole hippocampal formation were obtained with an APOTOME-ZEISS fluorescence microscope with a 10× objective using AxioVision software version 4.8 (Carl Zeiss, Oberkochen, Germany). The fluorescence signal of each marker was quantified in the *subiculum* of the hippocampal formation through image processing in Image J Fiji (NIH, USA) software version 1.54 f (Java 1.8.0_322) and expressed in µm^2^.

### 4.17. Statistical Analysis

The statistical analysis and visualization of the sample data were processed using GraphPad Prism 8 software version 8.0.2 (GraphPad Software, Inc., San Diego, CA, USA). The statistical tests for the comparison of groups were performed as follows. The one-way repeated measures Analysis of Variance (ANOVA) was used to analyze latencies from training in the MWM. The one-way ANOVA and Tukey’s post hoc analysis were used to analyze data on MWM, such as the latencies on day four of training and latencies of the retention, as well as data on the NOR task. The chi-square test was used for the statistical analysis of search strategies. The Kruskal–Wallis test and Dunn’s post hoc analysis were used for the statistical analysis of platform crossings in the MWM and data from assays of ACh metabolism and oxidative stress. The U-Mann Whitney test was applied for statistical analysis of data from immunofluorescent labeling. A *p*-value < 0.05 was considered significant.

## Figures and Tables

**Figure 1 ijms-26-01301-f001:**
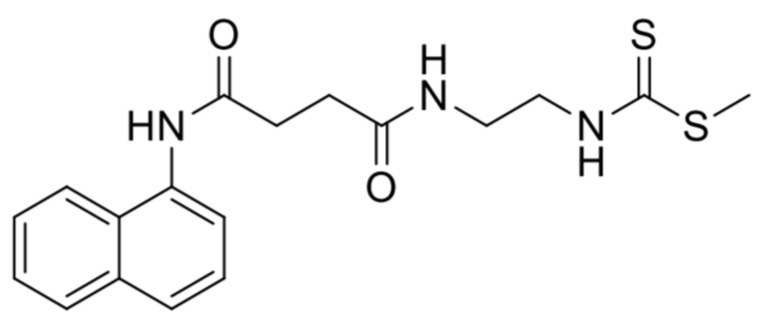
Chemical structure of CNEURO-201.

**Figure 2 ijms-26-01301-f002:**
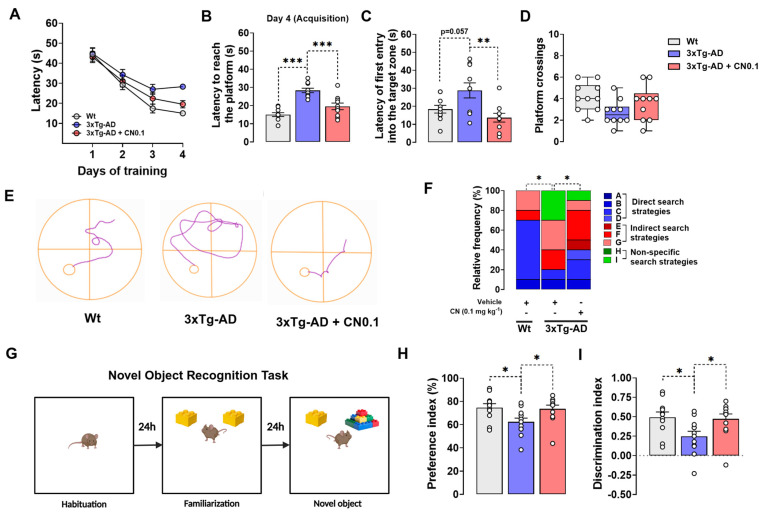
CNEURO-201 effect on cognitive performance. (**A**–**F**) Mice’s performance in the Morris water maze task. (**A**) Latencies to reach the platform along the training. (**B**) Latencies on day four of training. (**C**) Latencies for first entry into the target zone during the retention. (**D**) Number of platform crossings in the retention phase. (**E**) Representative images of the search strategies performed for each group of mice after the first entry into the target zone during the retention. (**F**) Stacked bars graph of the relative frequencies in percentage for each type of search strategy to reach the target zone during retention. (**G**) Illustrative representation of the novel object recognition (NOR) task used to evaluate the episodic memory of mice. (**H**) Preference index in percentage for the novel object and (**I**) Discrimination indexes obtained by each group of mice in the NOR task. Data on latencies, preference, and discrimination indexes are presented as mean ± s.e.m. Data on the number of crossings are presented as median and quartiles. Asterisks indicate significant differences (**p* < 0.05; ***p* < 0.01, ****p* < 0.001). MWM, n = 10 per group. NOR task: Wt, n = 11; *3xTg*-AD vehicle, n = 12; *3xTg*-AD CN0.1, n = 12. Non-transgenic mice (Wt) mice were treated with vehicle, whereas *3xTg*-AD mice received vehicle or 0.1 mg kg^−1^ CNEURO-201 (CN0.1).

**Figure 3 ijms-26-01301-f003:**
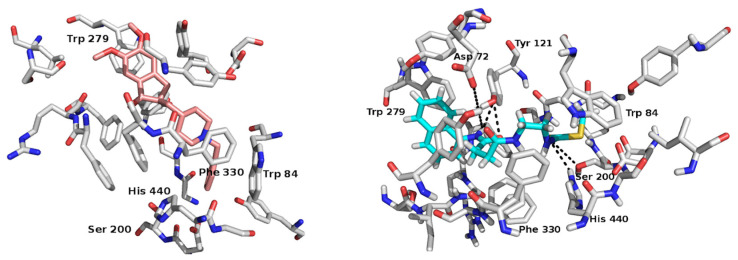
In silico analysis for the interaction between Donepezil acetylcholinesterase and CNEURO-201-acetylcholinesterase complexes. Ligand-AChE complexes (PDB code 1EVE) were obtained in molecular docking simulation with Donepezil and CNEURO-201. All ligands show interactions with amino acids of the peripheral anionic site (PAS) (Trp84, Trp279, and Phe330) and catalytic active site (CAS) (His440 and Ser200). The structure of AChE is shown in white. The compounds Donepezil and CNEURO-201 are shown in light pink and blue, respectively. Nitrogen atoms are represented in blue, oxygen atoms in red, and H-bonds in dashed lines.

**Figure 4 ijms-26-01301-f004:**
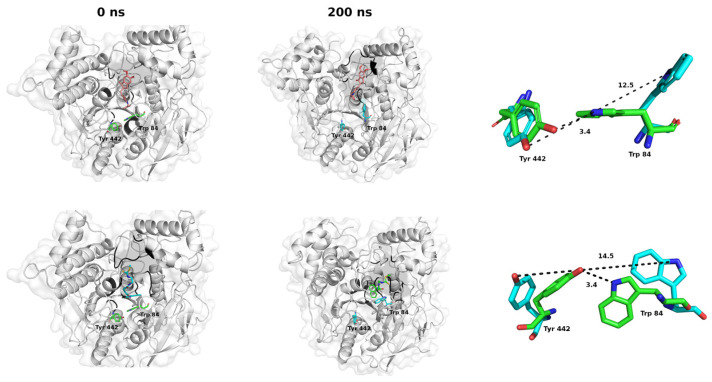
Molecular dynamics of the CNEURO-201-AChE system. A representative snapshot of the conformational evolution of CNEURO-201-AChE and Donepezil-AChE complexes at 0 and 200 ns, pH 7.4 and 310 K. Catalytic site amino acids (black). On the right, the relative orientation of the amino acids Tyr442 and Trp84 during molecular dynamics simulation studies is shown. Green represents the position at the initial time, and light blue represents the end of the 200 ns of simulation. Red represents oxygen atoms, and blue represents nitrogen atoms. Variation of the distance between Tyr442 Oζ-Trp84 Nε1 at 0 and 200 ns of simulation time. In dashed lines, the distances between Tyr442 Oζ-Trp84 Nε1.1.

**Figure 5 ijms-26-01301-f005:**
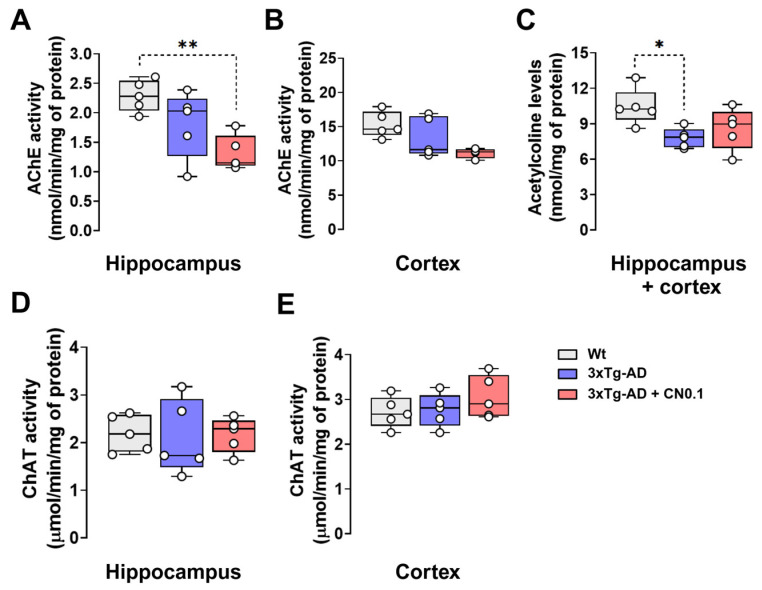
CNEURO-201 effects on brain acetylcholine metabolism. (**A**,**B**) Acetylcholinesterase (AChE) activities (nmol/min) in the *hippocampus* (**A**) and cortex (**B**) (n = 4–5). (**C**) Acetylcholine concentrations (nmol) of pooled samples of *hippocampus* and cortex (n = 5). (**D**,**E**) Choline acetyltransferase activities (µmol/min) determined in the *hippocampus* (**D**) and cortex (**E**) (n = 5). All data were normalized for milligrams of protein and are presented as median and quartiles. Asterisks indicate significant differences (**p* < 0.05; ***p* < 0.01). Non-transgenic mice (Wt) mice were treated with vehicle, whereas *3xTg*-AD mice received vehicle or 0.1 mg kg-1 CNEURO-201 (CN0.1).

**Figure 6 ijms-26-01301-f006:**
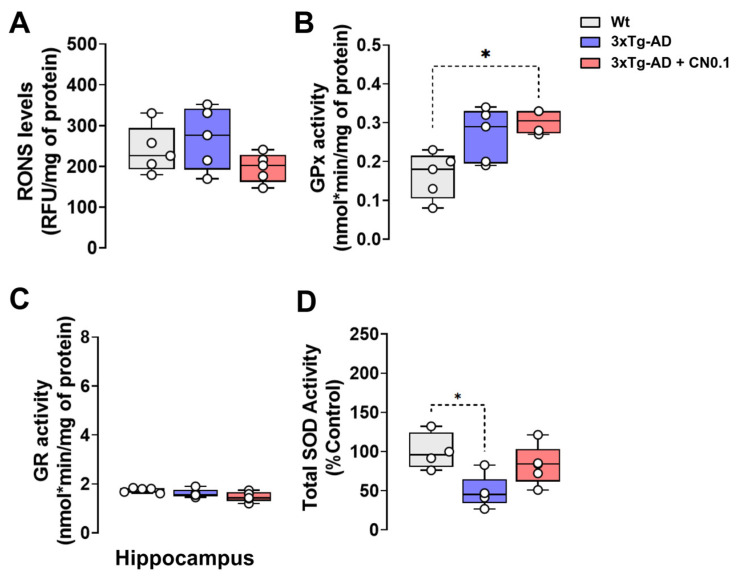
CNEURO-201 effect on oxidative stress and enzymatic activity of redox-defense related proteins in the hippocampus of mice. (**A**) Change in relative fluorescent units (RFU) of 2,7-dichlorofluorescein oxidation by free radicals (n = 5). (**B**) Glutathione peroxidase (GPx) activity in nmol*min (n = 4–5). (**C**) Glutathione reductase (GR) activity in nmol*min (n = 5). (**D**) Percentage of control results for total superoxide dismutase (SOD) inhibitory activity (n = 4–5). All data are presented as median and quartiles. Asterisks indicate significant differences (**p* < 0.05). Data are normalized by mg of protein. Non-transgenic mice (Wt) mice were treated with vehicle, whereas 3xTg-AD mice received vehicle or 0.1 mg kg^−1^ CNEURO-201 (CN0.1).

**Figure 7 ijms-26-01301-f007:**
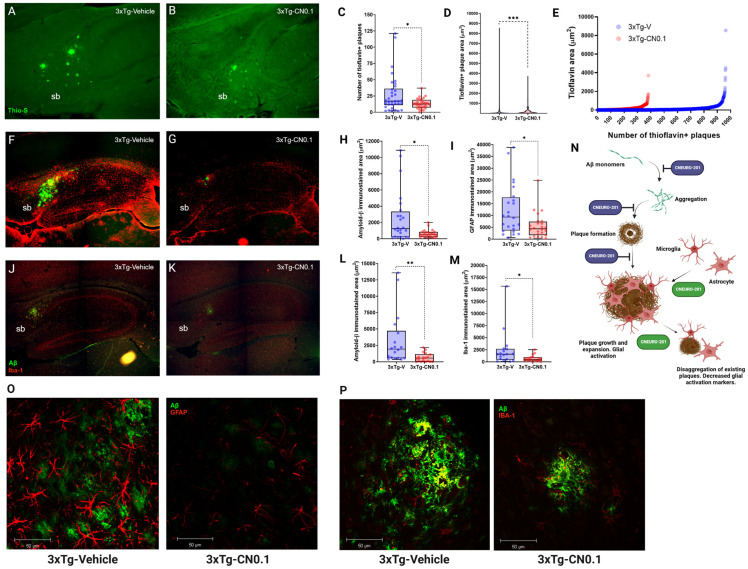
CNEURO-201 effect on amyloid-β plaques and glial cells in the subiculum from the hippocampal formation of *3xTg*-AD mice. (**A**,**B**) 25× representative images of the subiculum (sb) after thioflavin staining showing fluorescence (green) of dense amyloid-β (Aβ) plaques in the *3xTg*-AD mice treated with (**A**) vehicle (V) and (**B**) 0.1 mg kg^−1^ CNEURO-201 (CN0.1). (**C**) Number of thioflavin-positive plaques in the sb. (**D**) Violin plots showing thioflavin-positive plaque area in µm^2^. (**E**) 2D graph of number vs. area of thioflavin-positive plaques in the sb of *3xTg*-AD mice treated with the vehicle and CN0.1. (**F**,**G**) 10× representative images of the mouse hippocampal formation showing Aβ (green) and GFAP (red) immunofluorescent labeling of *3xTg*-AD mice treated with (**F**) vehicle and (**G**) CN0.1. (**H**) Aβ immunolabeled area in µm^2^ quantified in the sb (n = 4). (**I**) GFAP immunolabeled area in µm^2^ quantified in the sb (n = 4). (**J**,**K**) 10× representative images of the mouse hippocampal formation showing Aβ (green) and Iba-1 (red) immunofluorescent labeling of *3xTg*-AD mice treated with (**J**) vehicle and (**K**) CNEURO-201. (**L**) Aβ immunolabeled area in µm^2^ quantified in the sb (double immunofluorescence with Iba-1) (n = 3). (**M**) Iba-1 immunolabeled area in µm^2^ quantified in the sb (n = 3). (**N**) Schematic representation of possible mechanisms involved in the alleviation of amyloid pathology in Alzheimer’s disease by CNEURO-201 (legend in purple and green circles indicates inhibition or activation, respectively. Image created in BioRender.com). Figures in the bottom are high-resolution 40× images obtained with a confocal microscope showing fluorescence of GFAP+ astrocytes (red) and Aβ (green) (**O**), and Iba-1+ microglia (red) and Aβ (green) in the sb of 3xTg-AD mice treated with vehicle and CN0.1 (**P**). Note that reduction in Aβ plaque size by CN treatment results in changes of astrocytic morphology, as well as the presence of Iba-1 positive cells in the plaque area. All data are presented as median and quartiles. Asterisks denote significant differences between groups (**p* < 0.05; ***p* < 0.01; ****p* < 0.001).

## Data Availability

The raw data supporting the conclusions of this article will be made available by the authors on request.

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
