# Peer review of "CNEURO-201, an Anti-amyloidogenic Agent and σ1-Receptor Agonist, Improves Cognition in the 3xTg Mouse Model of Alzheimer’s Disease by Multiple Actions in the Pathology"

_ijms, 2025, doi:10.3390/ijms26031301_

Round 1
Reviewer 1 Report
Comments and Suggestions for Authors
The study “ CNEURO-201, an anti-amyloidogenic agent and σ1-receptor ag-2 onist improves cognition in the 3xTg mouse model of Alz-3 heimer’s disease by multiple actions in the pathology” is interesting, it aims to develop a a treatment for Alzheimer's disease. This topic addresses a public health problem and has great potential to contribute to knowledge, however, has issues that need to be clarified.
-Anti-amyloidogenic agent refers to inhibition of amyloid synthesis or amyloid plaque formation? Clarify this point in the introduction or correct the title and the whole text.
Fig. 6. It is necessary to add high resolution immunohistochemistry images with the 40x objective so that the reader can appreciate the morphology of the amyloid aggregates and the morphology of microglia and astrocytes (Iba-1 and GFAP). Otherwise this figure lacks the necessary scientific rigor.
Author Response
Thank you very much for taking the time to review this manuscript. Please find the detailed responses below and the corresponding revisions/corrections highlighted/in track changes in the re-submitted files. We appreciate the observations made which have contributed greatly to improving this manuscript.
Comment 1: Anti-amyloidogenic agent refers to inhibition of amyloid synthesis or amyloid plaque formation? Clarify this point in the introduction or correct the title and the whole text.
Response 1: Thank you for pointing this out. The anti-amyloidogenic properties of CNEURO-201 (CN) are indeed related with its capacity to inhibit Aβ aggregation. The same term has been used in previous studies reported by our group (Rivera-Marrero et al., 2020; García-Pupo et al., 2024). However, we understand that the term “amyloidogenic” is typically used to describe the Aβ synthesis pathway that led to the production of Aβ species with higher propensity to aggregate. To avoid confusion, we have clarified the above mentioned as follows:
Lines 98-99: “…(σ1R) agonist and anti-amyloidogenic agent (i.e., it impedes aggregation and facilitates disaggregation of Aβ25-35 aggregates) with the greatest…”
Lines 525-526: “between the inhibition of Aβ aggregation (anti-amyloidogenic property) and σ1R agonist features that allow the activation of natural homeostatic mechanisms in the brain tissue…”
Comment 2: Fig. 6. It is necessary to add high resolution immunohistochemistry images with the 40x objective so that the reader can appreciate the morphology of the amyloid aggregates and the morphology of microglia and astrocytes (Iba-1 and GFAP). Otherwise this figure lacks the necessary scientific rigor.
Response 2: Agree. We have, accordingly, added high-resolution representative 40x images for Aβ/GFAP and Aβ/Iba-1 double immunofluorescence in the subiculum of the mice groups (Panels O and P). Therefore, the figure and its caption as well as text in section 2.6 have been modified and updated in the manuscript.
Figure 7. CNEURO-201 effect on amyloid-β plaques and glial cells in the subiculum from the hippocampal formation of 3xTg-AD mice. (A, B) 25x representative images of the subiculum (sb) after thioflavin staining showing fluorescence (green) of dense amyloid-β (Aβ) plaques in the 3xTg-AD mice treated with (A) vehicle (V) and (B) 0.1 mg kg-1 CNEURO-201 (CN0.1). C) Number of thioflavin-positive plaques in the sb. D) Violin plots showing thioflavin-positive plaque area in µm2. E) 2D graph of number vs area of thioflavin-positive plaques in the sb of 3xTg-AD mice treated the vehicle and CN0.1. (F, G) 10x representative images of the mouse hippocampal formation showing Aβ (green) and GFAP (red) immunofluorescent labeling of 3xTg-AD mice treated with (F) vehicle and (G) CN0.1. H) Aβ immunolabeled area in µm2 quantified in the sb (n = 4). I) GFAP immunolabeled area in µm2 quantified in the sb (n = 4). (J, K) 10x representative images of the mouse hippocampal formation showing Aβ (green) and Iba-1 (red) immunofluorescent labeling of 3xTg-AD mice treated with (J) vehicle and (K) CNEURO-201. L) Aβ immunolabeled area in µm2 quantified in the sb (double immunofluorescence with Iba-1) (n = 3). M) Iba-1 immunolabeled area in µm2 quantified in the sb (n =3). N) Schematic representation of possible mechanisms involved in the alleviation of amyloid pathology in Alzheimer’s disease by CNEURO-201 (legend in purple and green circles indicate inhibition or activation, respectively). Figures in the bottom are high-resolution representative 40x images obtained with confocal microscope showing fluorescence of GFAP+ astrocytes (red) and Aβ (green) (O) and Iba-1+ microglia (red) and Aβ (green) in the sb of 3xTg-AD mice treated with vehicle and CN0.1. Note that reduction in Aβ plaque size by CN treatment results in changes of astrocytic morphology, as well as the presence of Iba-1 positive cells in the plaque area. All data are presented as median and quartiles. Asterisks denote significant differences between groups (*p < 0.05; **p < 0.01; ***p < 0.001).
Please see the attachment

Reviewer 2 Report
Comments and Suggestions for Authors
In this study, Humberto et al. present CNEURO-201 as an agent capable of disaggregating preformed amyloid aggregates, modulating acetylcholine metabolism, influencing the enzymatic activity of proteins involved in the redox system, and rescuing cognitive impairment. These effects were investigated through in silico, ex vivo, and in vivo studies. I would like to suggest the following revisions:
1. The current arrangement of the results is somewhat unclear. I recommend reordering the sections for better flow and logic. Specifically, the ex vivo results should precede the in vivo data for clarity.
2. manuscript uses 0.1 CN directly, but the reason for this particular dosage is not explained. Are there dose-dependent experiments that support this choice?
3. The authors should discuss and compare the dosage of CN used in this study with other similar studies, particularly to highlight its relatively low dose.
4. The authors conclude that no significant changes in ChAT activity were observed in the hippocampus or cortex. This finding warrants further exploration and discussion to understand the underlying implications.
5. To my knowledge, Aβ deposits in the frontal cortex of 3xTg mice by 6 months, primarily in layers 4 and 5, while tau tangles begin to form in CA1 neurons of the hippocampus at around 12 months. Given that this study appears to focus on tau pathology, the timeline and area selected seems more suited for studying tau rather than Aβ. Additionally, both tau and Aβ are thioflavin-positive. The authors should provide a clear explanation for this discrepancy.
Author Response
REVIEWER 2
Thank you very much for taking the time to review this manuscript. Please find the detailed responses below and the corresponding revisions/corrections highlighted/in track changes in the re-submitted files. We appreciate the observations made which have contributed greatly to improving this manuscript.
Comment 1: The current arrangement of the results is somewhat unclear. I recommend reordering the sections for better flow and logic. Specifically, the ex vivo results should precede the in vivo data for clarity.
Response 1: We appreciate your recommendation. However, the current order of sections was chosen based on the objectives of the research. First, we attempted to demonstrate that 0.1 mg/kg CNEURO-201(CN) treatment would improve cognitive function in 3xTg-AD mice (Section 2.1). Afterwards, we demonstrated through different techniques and analysis the effects of CN on several markers associated with the disease (cholinergic dysfunction, oxidative stress and glial activation) which correspond to ex vivo analysis performed in brain tissue of mice, except for in silico studies between CN and AChE in the sections 2.2 and 2.3.
Comment 2: manuscript uses 0.1 CN directly, but the reason for this particular dosage is not explained. Are there dose-dependent experiments that support this choice?
Response 2: Thank you for pointing this out. The dose of 0.1 mg/kg CN used in this study was chosen based on previous research by García-Pupo et al., 2024 and coworkers. In this work, it was found that CN doses of 0.03 and 0.1 mg/kg were more effective against cognitive decline in a mouse model of AD. The mentioned evidence is cited in the following lines of the introduction:
Lines 99-100 “…with the greatest effects preventing cognitive deficits in the dizocilpine-induced mouse model of AD at very low doses (0.03-0.1 mg kg-1) [31].”
Lines 104-105 “Bearing in mind that the efficacy of CN has been observed at lower doses [31] …”
Also, in the lines 641-643 of the methodology: “The dose of CN was chosen because in previous studies it was shown to be one of the most effective in preventing cognitive decline in a mouse model of AD [31].”
Comment 3: The authors should discuss and compare the dosage of CN used in this study with other similar studies, particularly to highlight its relatively low dose.
Response 3: CN is a novel molecule currently investigated at preclinical level. Our previous research has revealed that CN acts as an anti-Aβ agent as well as σ1R agonist. CN is not the first compound with σ1R agonist tested as a treatment for Alzheimer’s disease, for example, other types of these drugs like ANAVEX-2-73 and PRE-084 have been studied extensively. Nonetheless, CN is unique in his class because its interaction with Aβ aggregates and anti-aggregating and disaggregating properties against this species have been demonstrated previously (Rivera-Marrero et al., 2020; García-Pupo et al., 2024) and corroborated at a low dose of 0.1 mg/kg in the present manuscript. Therefore, we could discuss this information as follows:
Lines 516-530: “…This is also consistent with recent evidence demonstrating that 0.1 and even 0.03 mg kg-1 CN improves memory impairment induced by dizocilpine [31]. This was interesting, since drugs with σ1R agonistic activity like ANAVEX 2-73 showed greater improvements in memory with higher doses (0.3 and 1.0 mg/kg) in the same animal model [66]. Moreover, our data agrees with previous evidence supporting the potential role of σ1R agonists preventing memory deficits associated with Aβ [66,67]. However, CN stands out from other compounds with σ1R agonistic activity for its direct interaction of CN with Aβ aggregates and further disaggregation of this species. Therefore, we determined that the therapeutic effects of low doses of CN in AD stem from an orchestrated activity between the inhibition of Aβ aggregation (anti-amyloidogenic property) and σ1R agonist features that allow the activation of natural homeostatic mechanisms in the brain tissue. Thus, CN can display a notable capability to improve other hallmarks of AD pathology, such as increased oxidative and cholinergic dysfunction and disruption of glial processes involved in repair and anti-inflammatory responses.”
Comment 4: The authors conclude that no significant changes in ChAT activity were observed in the hippocampus or cortex. This finding warrants further exploration and discussion to understand the underlying implications.
Response 4: We agree with this comment. For that reason, we have discussed some plausible explanations in relation to this issue as described in:
Lines 583-595: “a plausible explanation may arise from the direct interaction that exists between AChE and Aβ. Previous studies have shown that AChE coexists with Aβ deposits and that the β-amyloid peptide can influence AChE levels and activity [75-77]. Thus, it seems logical to assume that as CN reduces Aβ burden, this may reduce AChE activity, which is of particular significance given that ChAT activity remained unaltered in both vehi-cle- and CN-treated 3xTg-AD mice. This suggests that CN acts as a positive modulator of the ACh system. This result is related to a study indicating that ACh release in the brain is increased via Ca2+ signaling following positive σ1R stimulation [70,71]. In this context, the higher ACh concentrations found in the present study could be indicative of an increase and preservation of synaptic ACh stores, rather than an increase in the synthesis of this neurotransmitter by ChAT. In light of these observations, it can be as-sumed that the σ1R agonist activity of CN also enhances ACh signaling.”
Comment 5: To my knowledge, Aβ deposits in the frontal cortex of 3xTg mice by 6 months, primarily in layers 4 and 5, while tau tangles begin to form in CA1 neurons of the hippocampus at around 12 months. Given that this study appears to focus on tau pathology, the timeline and area selected seems more suited for studying tau rather than Aβ. Additionally, both tau and Aβ are thioflavin-positive. The authors should provide a clear explanation for this discrepancy.
Response 5: Agree. Tau pathology also plays a crucial role in the physiopathology of AD in the 3xTg mouse model. However, there is confusion regarding the comment “this study appears to focus on tau pathology” since our study is focused only on Aβ due to the anti-aggregating property of CN. In this sense and based on the more recent characterization of 3xTg-AD mouse model by its developers (Javonillo et al., 2021), which shows that Aβ deposits in this transgenic strain become evident after 12 months old, we subjected 12-14 months-old 3xTg-AD and Wt mice to the treatment with CN for 8 weeks. At the end of the treatment, 14-16 months-old 3xTg-AD mice exhibited well stablished Aβ deposits mainly in the subiculum which were fluorescent after thioflavin-S staining. These plaques in this region of the hippocampal formation are primary composed by Aβ as we demonstrated in our immunofluorescence assays with BAM-10 antibody. We agree with the comment “both tau and Aβ are thioflavin-positive” and that tau pathology in the 3xTg-AD typically appears at later age. However, as you mentioned, Tau neurofibrillary tangles positive to thioflavine are observed in the CA1 subregion (Shin et al., 2021). Therefore, only thioflavin-positive plaques in the subiculum were analyzed to avoid bias by the possible presence of thioflavine-positive tau aggregates.
The later explanation can be found in the lines 398-404 of the 2.6 section of results and modificated as follows: “To this end, we designed an experiment in which we administered 0.1 mg kg-1 CN for eight weeks to aged 3xTg-AD mice (12-14 months of age; n = 3-4), which are known to exhibit the presence of plaques [29,49]. We conducted the study using histological staining of brain sections with the fluorescent dye thioflavin-S, which stains dense plaques and some tau aggregates in the CA1 subregion [50]. To avoid bias by the possible presence of thioflavin-positive tau aggregates, only the subiculum of the hippocampal formation was selected as the structure of interest due to its higher Aβ plaque abundance [29,49].”
In addition, due to the relevance of Tau pathology in AD, we are considering attending this issue in a further study. Please let us know any other questions or comments you may have.
Please see the attachment

Reviewer 3 Report
Comments and Suggestions for Authors
This is a high-quality paper studying CNEURO-201 (CN), a novel pharmaceutical agent with dual activity as an anti-amyloid-β (Aβ) agent and σ1 receptor agonist, as a multitarget drug candidate against Alzheimer´s disease (AD). Indeed, the effect of CN (1 mg/kg/day) on acetylcholine metabolism, oxidative stress, glial activation, and cognitive abilities, altered in AD, in 3xTg-AD mice was investigated after 8-week treatment. The paper is interesting and scientifically important in the field of the development of novel AD therapeutics, which is surely an actual scientific direction. The paper is well-written and can be understood easily. I have a few suggestions to additionally improve the significance of this study.
1. Title: I recommend the authors mention the terms associated with cholinergic functions, oxidative stress, and/or glial activation, which are the objectives of this study. To avoid too long titles, I suggest removing “an anti-amyloid-β (Aβ) 18 agent and σ1 receptor agonist”.
2. Introduction, the last paragraph: The authors provided relevant references on CN actions as a promising AD drug candidate. But in this paragraph, the authors should emphasize the novelty of this study (such as studying the effect on the investigated pathways for the first time) and its scientific impact on developing and exploring novel AD drugs. Then, the authors stated: “but its mechanism of action remains under investigation”. The action of CN as an anti-amyloid-β (Aβ) agent and σ1 receptor agonist is one of the mechanisms of its pharmacological action to improve AD-related symptoms. Thus, I would revise this sentence. Then, the authors state “this novel drug” – this is still a novel promising drug candidate to treat AD; it is not still approved and applied in clinical practice.
3. I recommend presenting the structure of CN after the last paragraph of the Introduction.
4. Materials and methods: The authors provided that AChE and ChAT activities were expressed as nmol/min. Please specify nmol of which compound.
5. Results, Figure 4: ChE and ChAT activities in the y-axis are expressed as nmol*min-it should be in nmol/min. Why was Ach level determined in pooled samples of hippocampus and cortex (AChE and ChAT activities were determined in hippocampus and cortex separately? Then, CN induced a significant decrease in AChE in the hippocampus compared to the control, whereas it was not observed in the cortex. What could be the reason for this?
6. Results: In silico analysis showed that CN interacts with Ser200 and His440 located in CAS of AChE. As the substrate binds directly to Ser200 of CAS, this suggests that CN might be a competitive, even irreversible AChE inhibitor. To reveal the nature of CN-AChE, I suggest the authors perform enzymatic kinetic analysis under in vitro conditions, with isolated and purified AChE. These experiments would complete and confirm the in silico analysis. In addition, circular dichroism (CD) spectroscopy might elucidate AChE-CN interaction at the molecular level. I suggest that these experiments be performed in another study (paper) to continue exploring the mechanism of CN action as a potential anti-AD drug.
7. Many reports on AD-related studies have been published in the last 5 years. Thus, the authors might replace older references with more recent ones.
Author Response
Reviewer 3
Thank you very much for taking the time to review this manuscript. Please find the detailed responses below and the corresponding revisions/corrections highlighted/in track changes in the re-submitted files. We appreciate the observations made which have contributed greatly to improving this manuscript.
Comment 1: Title: I recommend the authors mention the terms associated with cholinergic functions, oxidative stress, and/or glial activation, which are the objectives of this study. To avoid too long titles, I suggest removing “an anti-amyloid-β (Aβ) 18 agent and σ1 receptor agonist.
Response 1: Thank you for your kind suggestion. However, we would like to conserve the original title since it points out the dual properties of our molecule which are part of the novelty of CN and related with the markers that are the objectives of the study.
Comment 2: Introduction, the last paragraph: The authors provided relevant references on CN actions as a promising AD drug candidate. But in this paragraph, the authors should emphasize the novelty of this study (such as studying the effect on the investigated pathways for the first time) and its scientific impact on developing and exploring novel AD drugs. Then, the authors stated: “but its mechanism of action remains under investigation”. The action of CN as an anti-amyloid-β (Aβ) agent and σ1 receptor agonist is one of the mechanisms of its pharmacological action to improve AD-related symptoms. Thus, I would revise this sentence. Then, the authors state “this novel drug” – this is still a novel promising drug candidate to treat AD; it is not still approved and applied in clinical practice.
Response 2: Thank you for pointing this out. We consider your observations about our last paragraph in the introduction to be important and appropriate for modification. We have rewritten the text as follows:
Lines 101-110: “Preclinical studies on CN indicate a strong potential as a candidate to treat AD. However, despite the action of CN as an anti-Aβ agent and σ1 receptor agonist is one of the mechanisms of its pharmacological action to improve AD-related symptoms, the impact of this effects in associated markers of the pathology remains unclear. Moreover, bearing in mind that the efficacy of CN has been observed at lower doses [31], the aim of this study was to perform an oral intervention in the 3xTg-AD mice with 0.1 mg kg-1 CN, to investigate its effects on learning and memory and relevant biomarkers as-sociated with other altered pathways beyond Aβ. It is therefore anticipated that treatment with 0.1 mg kg-1 CN will exert a beneficial influence on cholinergic dysfunction, glial activation, and oxidative stress in memory substrate brain regions of 3xTg-AD mice.”
Comment 3: I recommend presenting the structure of CN after the last paragraph of the Introduction.
Response 3: Thank you for your kind suggestion. We have included a figure with the structure of CN as you suggested. Therefore, the number of figures has been updated.
Comment 4: Materials and methods: The authors provided that AChE and ChAT activities were expressed as nmol/min. Please specify nmol of which compound.
Response 4: Agree. Therefore, we have mentioned this information as follows:
Lines 720-721: “The AChE activity was expressed as nmol of acetylthiocholine catalyzed/min and normalized per mg of protein.”
Lines 751-752: “The ChAT activity was expressed as µmol of acetylcholine formed/min and normalized per mg of protein”
Comment 5: Results, Figure 4: AChE and ChAT activities in the y-axis are expressed as nmol*min-it should be in nmol/min. Why was Ach level determined in pooled samples of hippocampus and cortex (AChE and ChAT activities were determined in hippocampus and cortex separately? Then, CN induced a significant decrease in AChE in the hippocampus compared to the control, whereas it was not observed in the cortex. What could be the reason for this?
Response 5: We have modified the figure 4 (now figure 5) and its figure caption as follows:
“Figure 5. CNEURO-201 effects on brain acetylcholine metabolism. (A, B) Acetylcholinesterase (AChE) activities (nmol/min) in the hippocampus (A) and cortex (B) (n = 4-5). C) Acetylcholine concentrations (nmol) of pooled samples of hippocampus and cortex (n = 5). (D, E) Choline acetyl-transferase activities (µmol/min) determined in the hippocampus (D) and cortex (E) (n = 5). All data were normalized for milligrams of protein and are presented as median and quartiles. Asterisks indicate significant differences (*p < 0.05; **p < 0.01). Non-transgenic mice (Wt) mice were treated with vehicle, whereas 3xTg-AD mice received vehicle or 0.1 mg kg-1 CNEURO-201 (CN0.1).”
Regarding the use of pooled samples of the hippocampus and cortex for the acetylcholine measurements, this was considering two aspects: 1) the similarities in the reductions of AChE activity in the hippocampus and cortex (~42% decrease compared with Wt). Therefore, we expected the changes would be modified in the same direction (i.e, increase) in these brain structures, particularly due to the absence of changes in the ChAT activities. 2) to ensure the detection of optimal concentrations of acetylcholine for the analysis, since there was no previous evidence available for this methodology in the brain of 3xTg-AD mice.
Regarding the absent significant changes in the AChE activity of cortex compared with the hippocampus. Our results suggest that CN could influence AChE activity among the different brain structures of the brain, however, for the case of 3xTg-AD mice, a plausible explanation could rely on the progression of Aβ pathology in this strain, which begin in the hippocampus and is extended to the cortex at later age. Indeed, AChE activity has been suggested to be decreased with age in human patients with Alzheimer’s disease and the 3xTg-AD mice (Campanari et al, 2016; Várkonyi et al, 2022), which could be a consequence of the loss of cholinergic fibers (Pérez et al, 2011; Várkonyi et al, 2022). However, this issue remains unclear and deserves further investigation.
Comment 6: Results: In silico analysis showed that CN interacts with Ser200 and His440 located in CAS of AChE. As the substrate binds directly to Ser200 of CAS, this suggests that CN might be a competitive, even irreversible AChE inhibitor. To reveal the nature of CN-AChE, I suggest the authors perform enzymatic kinetic analysis under in vitro conditions, with isolated and purified AChE. These experiments would complete and confirm the in silico analysis. In addition, circular dichroism (CD) spectroscopy might elucidate AChE-CN interaction at the molecular level. I suggest that these experiments be performed in another study (paper) to continue exploring the mechanism of CN action as a potential anti-AD drug.
Response 6: We agree with your commentary, and we are currently working on an experimental design to address this question for a future paper.
Comment 7: Many reports on AD-related studies have been published in the last 5 years. Thus, the authors might replace older references with more recent ones.
Response 7: We agree with your commentary. We have replaced references 2, 3, 6, 43, 45, 48, and 54. It is important to highlight that relevant evidence related to cholinergic dysfunction is still older, but they represent important insights in the AD field.
Please see the attachment

Round 2
Reviewer 2 Report
Comments and Suggestions for Authors
Thanks for addressing my comments